# A Novel Deoxyribonuclease Low-Molecular-Weight Bacteriocin, Carocin S4, from *Pectobacterium carotovorum* subsp*. carotovorum*

**DOI:** 10.3390/microorganisms11071854

**Published:** 2023-07-22

**Authors:** Huang-Pin Wu, Reymund C. Derilo, Shih-Hao Hsu, Jia-Ming Hu, Duen-Yau Chuang

**Affiliations:** 1Division of Pulmonary, Critical Care and Sleep Medicine, Chang Gung Memorial Hospital, Keelung 33305, Taiwan; whanpyng@cgmh.org.tw; 2College of Medicine, Chang Gung University, Taoyuan 33302, Taiwan; 3Department of Chemistry, National Chung-Hsing University, Taichung City 40227, Taiwan; rcderilo@nvsu.edu.ph (R.C.D.); shinhaohsu@gmail.com (S.-H.H.); jamyhu2010@gmail.com (J.-M.H.); 4College of Teacher Education, Nueva Vizcaya State University-Bambang Campus, Bambang 3702, Philippines

**Keywords:** *Pectobacterium carovotorum* subsp*. carotovorum*, bacteriocin, Carocin, killer protein, immunity protein, low-molecular-weight bacteriocin

## Abstract

*Pectobacterium carotovorum* subsp*. carotovorum* (*Pcc*) is known to produce different types of bacteriocins, active protein substances that inhibit or kill related strains and are known to be induced by several factors. In this paper, we report the discovery, isolation, characterization, and functional analysis of Carocin S4, a novel low-molecular-weight bacteriocin (LMWB) from *Pcc*. A 2750 bp gene fragment was isolated from the chromosomal DNA of *Pcc* mutant strain rif-TO6, a rifampicin-resistant strain of TO6. The gene contains *caroS4K* and *caroS4I* within two open reading frames, which encode CaroS4K and CaroS4I, with molecular weights of about 90 kD and 10 kD, respectively. The unique characteristics of Carocin S4 were revealed after homology analysis with the previously discovered bacteriocins from *Pcc*. CaroS4K, which shares 23% and 85% homology with CaroS1K and CaroS3K, respectively, is also a deoxyribonuclease. However, unlike the two which can only hydrolyze genomic DNA, CaroS4K hydrolyzes both genomic and plasmid DNA. On the other hand, CaroS4K was found to be 90% homologous with CaroS2K but works differently in killing the target cell, as the latter is a ribonuclease. The optimal reaction temperature for CaroS4K to hydrolyze dsDNA is approximately 50 °C and requires the divalent metal ions Mg^2+^, Ca^2+^, and Zn^2+^ to catalyze its DNase activity. This study reveals another nuclease type of bacteriocin in *Pcc*, with CaroS4K and CaroS4I functioning as killer and immunity proteins, respectively.

## 1. Introduction

*Pectobacterium carotovorum* subsp*. carotovorum* (*Pcc*) is a pathogen which causes soft-rot disease in numerous plants worldwide, resulting in significant economic losses [1]. *Pcc* is suited to living in an environment of 28 to 32 °C and can metabolize the production of organic acids [2,3]. This strain can cause tissue decay or plant decay in agricultural and horticultural crops, reducing the economic value of plants [4]. *Pcc* strains do not directly invade the plant epidermis. Insect bites and wounds give the pathogens in the soil the opportunity to invade [5,6], causing the corruption of plant storage tissues [7], such as roots, bulbs, tubers, and other nutrient organs, as well as making the tips of plant leaves yellow and wither and, in severe cases, causing plant death and the production of H_2_S odors [8]. According to previous studies, several strains often produce antibiomass, called bacteriocins, that inhibit the growth of other related strains [9]. Bacteriocins are active protein substances that inhibit or kill related strains which are usually found in the soil and the human body and even in food [10].

Both Gram-positive and Gram-negative bacteria are known to produce bacteriocins [11]. Bacteriocins, such as those produced by *Escherichia coli*, can be categorized as RNase (e.g., colicins E3, E4, and E6), DNase (e.g., colicins E2, E7, E8, and E9), tRNase (e.g., colicins D and E5), and pore-forming colicins (e.g., colicins A, E1, Ia, and Ib) [12]. Usually, nuclease bacteriocins have a high amino acid sequence homology. Among the extensively studied bacteriocins are Nisin, produced by *Lactoccus lactis* (a Gram-positive bacterium), and Colicin, produced by *Pseudomonas aeruginosa* (a Gram-negative bacterium) [13,14]. The secretion of bacteriocins is mostly due to the threat to the survival of bacteria; factors such as temperature, humidity, ultraviolet light, pH, nutrient sources, chemicals, and other agents may induce their production [15,16].

A total of four bacteriocins secreted by several *Pcc* strains have already been discovered in our laboratory. These include Carocin S1 [17], Carocin S2 [18], Carocin S3 [1], and the newly discovered bacteriocin which is extensively discussed in this study, Carocin S4. This paper reports the discovery, cloning, expression, characterization, and functional analysis of the LMWB Carocin S4. It also describes how the novel bacteriocin differs from the previously discovered *Pcc* bacteriocins in terms of its killing activity. To date, research on Carocin is limited, and it remains unclear how Carocin identifies the target cells and how the killer protein delivers its killer domains. Hence, in this study, we purified the bacteriocin’s killer protein and explored its primary characteristics. The findings shed light on how the bacteriocin is able to uniquely hydrolyze DNA compared to the previously described bacteriocins from *Pcc* and how external factors, such as temperature and divalent metal ions, affect the killing activity of the bacteriocin. 

## 2. Materials and Methods

*Bacterial strains, media, and growth conditions.* The bacterial strains, plasmids, and primers used in this study are listed in Appendix A. *Pcc* strains were grown at 28 °C in a modified Luria–Bertani (LB) medium containing 5 g of sodium chloride per liter (half the recommended quantity of NaCl). *E. coli* strains (the cloning hosts) were cultured at 37 °C with rotary agitation at 125 rpm in LB broth. Quantities of 1% polypeptin, 0.2% yeast extract, 0.1% MgSO_4_ (pH 7.0), and 1.5% agar were added to the IFO-802 medium. Antibiotic concentrations used in *E. coli* selection—ampicillin, 50 g/mL; kanamycin, 50 g/mL; and rifampicin, 50 g/mL—were used to treat the *E. coli* and *Pcc* strains. A spectrophotometer was used to detect all bacterial growth densities at 595 nm (OD_595_).

*Bacterial conjugation.* Overnight cultures of *Pcc* (recipient) and *E. coli* (donor) were mixed and spread onto 0.22 μm membrane filters placed on LB agar media and incubated overnight at 28 °C [17,19]. The progeny after conjugation were appropriately diluted and cultivated on Modified Drigalski’s medium (with ampicillin and kanamycin (100 μg mL^−1^)) overnight at 28 °C. All isolates were placed on IFO-802 medium and tested for bacteriocins. Bacteriocins were assayed using the double-layer method, and *Pcc* SP33 was used as an indicator strain [20]. The cells were incubated for 12 h to form colonies, exposed to ultraviolet irradiation, incubated again for 12 h, treated with chloroform to kill the cells, and then covered with soft agar containing indicator cells. The bacteriocin production was indicated by a zone of inhibition of indicator cell (SP33) growth around the colony.

*Genetic engineering technique.* Previously described techniques were used to isolate the plasmids of *Pcc* [21,22] and *E. coli* [23]. Total DNA was isolated as previously described [22]. Oligonucleotide DNA primers were synthesized by MDE Bio Inc. (Taipei, Taiwan). Reagents were purchased from Takara Co. (Tokyo, Japan). Previously detailed protocols were utilized for the general polymerase chain reaction (PCR) [24] and thermal asymmetric interlaced PCR (TAIL-PCR) [25]. For TAIL-PCR, specific primers complementary to the respective sequences of Tn*5* (PR-1, PR-2, PR-3, PF-1, PF-2, and PF-3) were used. In addition, three arbitrary degenerate primers designated N-1, N-2, and N-3 were used.

TAIL-PCR products were sequenced using an ABI PRISM Dye Terminator Cycle Sequencing Ready Reaction kit (Applied Biosystems, Foster City, CA, USA). Cycle sequencing was carried out in a GeneAmp System 9600 thermocycler (Applied Biosystems). Sequencing was carried out according to the manufacturer’s protocol using an ABI 373S automated DNA sequencer 373S (Applied Biosystems).

Southern and colony hybridizations, probe labeling, and detection were performed using a DIG DNA Labeling and Detection kit (Boehringer Mannheim GmbH, Mannheim, Germany) as described by the manufacturer. Hybridization was performed overnight, and the membrane was washed according to the recommendations of the manufacturer.

DNA electrophoresis, restriction digest, ligation, and transformation procedures for *E. coli* were performed as previously described [24]. Plasmid DNA transformation for *Pcc* was performed using two previously described methods [26,27] following an incubation at 35 °C until the optical density (550 nm) of the culture was 0.40 to 0.55.

*Protein purification.* The transformant cells of BL21, harboring pETS4KI or pETS4I, were grown in 500 mL to an OD_595_ of 0.4 [28]. The cells were induced with isopropyl-β-D-thiogalactopyranoside (IPTG; final concentration, 0.1 mM; at 37 °C for 12 h). Subsequently, the cells were pelleted, and the pellets were sonicated (10 cycles of 9 s with 9 s intervals). BL21/pETS4KI pellets were subjected to ammonium sulfate precipitation (30–40%), resuspended in buffer A (30 mM NaCl and 20 mM Tris-Cl, pH 8.0), and applied to a Fractogel column (Merck, Rahway, NJ, USA). The fraction was eluted by a NaCl gradient (30 mM–1.4 M). After purification through a P-100 size-exclusion column (BioRad, Hercules, CA, USA), the CaroS4K fractions were pooled and concentrated using an Amicon centriprep-50 column (Millipore, Burlington, MA, USA) and dissolved in buffer A. BL21/pETS4I pellets were precipitated by ammonium sulfate (70–100%) and resuspended in buffer A. CaroS4I purification involved a similar chromatographic procedure using the Amicon centriprep-3 column (Millipore, Burlington, MA, USA). The concentration of protein was determined by the Bradford assay (Amresco, Solon, OH, USA). In determining the protein content, we used Coomassie^®^ Brilliant Blue G-250 (Bio-Rad protein Assay) as an indicator to analyze protein concentration with the Bradford assay.

*In vitro determination of Carocin S4 activity.* Overnight cultures of 95F3 were diluted (1:100) with LB medium and grown at 28 °C to a density of approximately 10^5^ CFU ml^−1^. The activity of increasing concentrations of Carocin S4 on cells in suspension incubated at 28 °C for 60 min was assessed. CaroS4I was pre-mixed with an equal molar ratio of CaroS4K. All reaction mixtures were spread onto LB agar plates and incubated at 28 °C for 24 h. The experiment was performed three times. The colonies growing on a series of plates were counted. 

To confirm DNase activity, 1 μg of genomic DNA from 95F3 in solution containing buffer A was incubated with or without Carocin S4 for various lengths of time (at 28 °C for 0, 60, 180, 300, 420, and 540 min) and at various temperatures (5 to 70 °C, with 5° increases). Moreover, the activity relationship between divalent metal ions (Ca^2+^, Mg^2+^, Cu^2+^, and Zn^2+^) and CaroS4K was tested. The samples were then subjected to electrophoresis on 1% agarose gel.

*ICP-MS of CaroS4K.* The purified CaroS4K solution underwent dialysis twice with secondary deionized water, was adjusted to a concentration of 5 μM, and was sent to the Guiyi Center of Tsinghua University for ICP-MS testing.

*Computer analysis of sequence data.* The DNA nucleotide sequences and the deduced amino acid sequences of Carocin S1, Colicin E1, Colicin E2, and Colicin E3 were compared using the BLAST and FASTA programs of the National Center for Biotechnology Information server. Sequence data were compared using DNASIS MAX 3.0 software (Hitachi, Tokyo, Japan).

## 3. Results

### 3.1. Selection of Bacteriocin-Producing Strains

Through a systematic bacteriocin activity test for all the available strains in our laboratory, the relationships between the bacteriocins produced by each strain and the indicator strain were cross-compared and listed in Appendix A. The results showed that the same strain exhibits different sizes, visibilities, and shapes of bacteriocins depending on the indicator strain (Figure 1). Growth inhibition is also specific. After conducting a series of tests, we selected bacteriocin-producing strains that can effectively inhibit the growth of most indicator strains and for which the inhibition zones are most obvious and easy to observe. Through careful examination, strain TO6 was selected for further analysis.

### 3.2. Isolation and Detection of Transposon Tn5 Mutants

We selected rif-TO6, which was from the rifampicin-resistant strain TO6, as the recipient of bacterial conjugation and *E. coli* (1830) with the kanamycin resistance gene transposon Tn5 as the donor for conjugative reproduction. Among 15,000 strains, 14 mutant strains that completed conjugation and blocked LMWB-producing genes were initially screened, and the mutant strains were named TT6-X (the first T stands for transposon and is followed by T6, which is an abbreviation of strain TO6, and the X indicates the order of discovery, expressed in Arabic numerals). To screen the conjugated *Pcc*, we used modified Drigalski’s medium with the antibiotics kanamycin and rifamycin. Subsequently, the bacteriocin assays [29,30] were performed to find the strains whose low-molecular-weight bacteriocin-related genes were successfully disrupted.

Subsequently, the position of the relevant gene blocked by Tn5 and the protein function expressed by the gene were investigated. We purified the genomic DNA of the mutant strains and restricted it with *EcoR*I, which confirmed the purification results. In addition, PCR technology was used to confirm whether the above mutant strains have the unique *npt*II fragment on transposon Tn5. The length of *npt*II is about 490 bp, and *E. coli* 1830 was used as the control group (Figure 2). The results showed that the genetic DNA of each mutant strain (TT6-1, TT6-3, TT6-5, TT6-6, TT6-13, and TT6-21) does have *npt*II gene fragments, proving that the LMWB-producing gene of the selected mutant strain was disrupted by Tn5.

### 3.3. Amplification of the Gene Blocked by Transposon Tn5 Using the TAIL-PCR Method 

We obtained the genomic DNA purified by the mutant strains above, following the studies of Yao-Guang et al. [31]. Two groups of specific primers were designed outwards and in opposite directions using a known Transposon Tn5 sequence (PF1, PF2, PF3, PR1, PR2, and PR3) in conjunction with random primers (N1, N2, and N3) which can amplify from the Tn5 sequence to unknown DNA fragments on both sides. Since both sets of primers are at the outer end of the Transposon Tn5 gene, these two DNA fragments will contain unknown genes in addition to part of the Tn5 gene that may have been blocked.

The unknown sequence fragments of the specific primer and random primer were amplified using TAIL-PCR. The results are shown in Appendix A with the first (1), second (2), and third (3) amplifications using the PR1 and N3, PR2 and N3, PR3 and N3 primer pairs, respectively. The locations of the DNA fragments indicated by the arrows in the figure are in accordance with our expected results, so we purified and sequenced the location of the third TAIL-PCR product. Since the tail sequence of the Tn5 gene is ATAGAGTCAG, it was compared to the sequence upstream of the sequence results to determine whether the PCR product was a DNA fragment of Tn5 [25].

Using the NCBI Basic Local Alignment Search Tool (BLAST Tool), the DNA sequence was investigated for the function of genes near the insertion point of the Tn5 gene. The results were organized by selecting those with higher functional homogeneity. The homogeneity analysis of the genes inserted into the different mutant strains of Transposon Tn5 were analyzed. The results were the basis for designing the primers for the subsequent steps. In order to obtain the full gene of the bacteriocin-producing proteins, we used the results of TAIL-PCR sequencing to design two sets of specific primers in different directions using TO6 genomic DNA as a template to increase the length of unknown fragments. We used the TT6-1 ATPase 990 bp and TT6-6 diguanylate cyclase 1074 bp total gene lengths. The 2750 bp gene from TT6-3, TT6-13, and TT6-21, which is suspected to be the gene that produces bacteriocins, was amplified.

### 3.4. Bacterial Gene Selection and Bacteriocin Activity Test

After obtaining multiple TAIL-PCR sequencing results, we used the DNAsis (Mac) software to simulate the two complete open reading frames. The first reading frame, ORF1, displayed an amino acid sequence of 25% homogeneity with respect to the colicin D-157 killer protein in the NCBI database. We presumed that they have related functions and named ORF1 the *caroS4K* gene, which has a length of 2484 bp. For mutant strains TT6-3 and TT6-13, Tn5 was inserted between *caroS4K* 1274 bp and 1275 bp, while, for the mutant strain TT6-21, Tn5 was inserted between *caroS4K* 1382 bp and 1383 bp. For the second reading frame, designated ORF2, no information was obtained from the NCBI database. We speculated that it is related to the function of immunity; thus, it was named the 270 bp long gene *caroS4I*. The *caroS4K* and *caroS4I* genes are jointly named *carocin S4*, with a total gene length of 2750 bp.

*Carocin S4* is similar to the *pyocin* produced by *P. aeruginosa* and *colicin* produced by *E. coli* [32]. Hence, we speculate that the *carocin S4* gene is the one responsible for the production of bacteriocin proteins in *Pcc*. Using TAG cloning technology, we used the *carocin S4* to form a pGS4KI construct which was further transformed into a competent cell, DH5α, a non-bacteriocin-producing strain. After the multiple screening and testing of 48 transformant strains, we selected a strain with a stable production of bacteriocins. We purified the mass of this transformed mutant and confirmed it by a Southern blotting experiment (Figure 3). The 3484 bp construct contains the *carocin S4* gene, and the results showed both the vector and insert signals, so the pGS4KI construct was sent for genome walking to confirm the complete *carocin S4* gene sequence. The *carocin S4* gene was then inserted into the vector pGEM-T Easy and was named pGS4KI, which contains the *caroS4K gene* with a length of 2484 bp and a *caroS4I* gene with a length of 270 bp. In addition, *caroS4I* was also attached to the carrier pGEM-T Easy, named pGS4I, containing the *caroS4I* gene which could normally show immune activity, free from Carocin S4 attack. Subsequently, bacteriocin activity tests were conducted to confirm the functions of the two genes, *caroS4K* producing killing proteins and c*aroS4I* producing immunity proteins.

### 3.5. Transcriptional Analysis and In Vivo Expression of the Carocin S4 Gene

We purified the RNA from the strains rif-TO6, TT6-13, and TT6-13/pGS4KI and took 1 μg of RNA from each for RT-PCR. We designed a set of primers, TO6R-sens14 and TO6-probes, located in the *caroS4K* gene, and another set of primers located in the *caroS4I* gene, TO6IR-sens17 and TO6IR-anti11, with the expectation of increasing DNA fragments of 1513 bp and 268 bp. Figure 4c shows that the control-group ribosomal RNA appears at 397 bp, which means that the RNA is functioning properly. In Figure 4a, the cDNA of the Tn5 mutant strain in lane 2 cannot be observed, but the cDNA of both the recovery strain rif-TO6 (lane 1) and the mutant strain (lane 3) were amplified (1513 bp), indicating that Tn5 did disrupt the *caroS4K* gene, causing the bacteriocin gene not to be transcribed into RNA. In Figure 4B, all tested RNAs could produce the 397 bp DNA fragments, so we speculate that Tn5 did not block the *caroS4I* gene; hence, the gene can transcribe RNA normally. The RT-PCR experiment is consistent with our expected results, and the mutant strain’s inability to secrete bacteriocin was indeed due to Tn5 disruption of *caroS4K*. Moreover, it was also found out that the ORF predicted position of the *carocin S4* gene was correct.

The CaroS4K amino acid sequence was cross-matched using the Basic Local Alignment Search Tool (BLAST) provided by the National Institutes of Health medical library database (NCBI). The results showed that CaroS4K’s amino acid sequence from the 1st to the 678th amino acid is 87% homogeneous with the S-type pyocin domain. It is also 90% homogeneous with the sequence from the 1st to the 678th amino acid in CaroS2K. Moreover, its sequence from the 463rd to 821st amino acid is 58% homogeneous with colicin/pyocin nuclease family proteins.

### 3.6. Purification and Characterization of the Carocin S4 Gene

*E. coli* BL21 recombinants, which were transformed with pET32a and pETS4KI-2, were used to produce large amounts of expression proteins. Quantitative activation induced BL21/pET32a and BL21/pETS4KI-2. A bacterial solution was taken to perform bacterium dissolution experiments, and supersonic oscillations were used to obtain crude protein fluid from the remaining BL21/pETS4I bacteria. Thereafter, it was run in 8% SDS-page analysis, and the results showed that BL21/pETS4KI-2 is induced by IPTG and is responsible for a large amount of CaroS4K production (Figure 5A). The purified protein contains two proteins, one with a molecular weight of approximately 90 kD (CaroS4K) and the other with a weight of approximately 10.2 kD (CaroS4I) (Figure 5B).

A similar method was employed for the overexpression and purification of CaroS4I (Figure 6), using pET32a and pETS4I. The crude protein fluid containing CaroS4I was run in 12% SDS-page for purification. The purified CaroS4I protein has a molecular weight of approximately 10.2 kD.

After purification, CaroS4K was further tested and characterized. The purified CaroS4K effectively inhibited the growth of the indicator bacterium 95F3. The findings also showed that the higher the concentration of CaroS4K, the more significant the antibacterial effect was. The minimum antibacterial concentration of CaroS4K (killing 95% of 95F3) is approximately 0.2 μg/mL. Furthermore, the purified CaroS4K was then sent to Tsinghua University for the determination of its metal ion contents using ICP-MS. Table 1 shows the main metal ions in CaroS4K. These ions include Mg^2+^, Ca^2+^, Cu^2+^, and Zn^2+^.

### 3.7. DNase Activity of CaroS4K with Varying Times and Temperatures

DNA isolated from pMCL200, pGEM-4Z, pBluescript, and pQE30 with and without the presence of CaroS4K was tested in 10 mM Ca^2+^ buffer (Appendix A). The results showed that CaroS4K hydrolyzed all four bodies and resulted in DNA fragments of open circular and linear forms. Hence, we hypothesized that CaroS4K is a DNase with the properties of DNA endogenous enzymes. 

We further tested the effect of time and temperature on its DNase activity. It was observed that the genomic DNA of 95F3 was hydrolyzed by CaroS4K over time into smaller DNA fragments (Appendix A). Thus, we believed that CaroS4K kills 95F3 by hydrolysis of genomic DNA. Similarly, CaroS4K was tested with the DNA isolated from pMCL200 (Figure 7). It was observed that part of the supercoiled form of pMCL200 was uncoiled by CaroS4K to an open circular form and hydrolyzed into a linear form of pMCL200 after an hour of exposure. From the results, we speculate that the process of DNA hydrolysis via CaroS4K is as follows. First, it uncoils the supercoiled DNA into an open circular form. Then, it hydrolyzes the open circular DNA into a linear form. Finally, it further hydrolyzes the linear DNA into smaller pieces of DNA. From these results, we believe that the process by which CaroS4K hydrolyzes the DNA is similar to those of other DNase bacteriocins, such as Colicin E2, Colicin E7, Colicin E8, and Colicin E9 [33]. We speculate that CaroS4K is a non-specific endonuclease.

We also determined the optimum temperature for CaroS4K activity. Figure 8 shows the results of the experiments. DNA from pMCL200 was tested with and without CaroS4K in 10 mM Ca^2+^ buffer at varying temperatures. At a reaction temperature of 35 °C to 55 °C, the linear form of pMCL200 was hydrolyzed by CaroS4K into smaller DNA fragments, resulting in the largest number of smaller DNA fragments at 50 °C. Thus, the optimal reaction temperature for CaroS4K DNase activity is approximately 50 °C. There was no significant reduction in the amount of the supercoiled form of pMCL200 when the reaction temperature rose beyond the optimal temperature.

### 3.8. The Activity Relationship between the Divalent Metal Ions Ca^2+^, Mg^2+^, Cu^2+^, and Zn^2+^ and CaroS4K

Varying the concentrations of the four divalent metal ions and their effects on CaroS4K activity was tested on pMCL200. The reaction was carried out for 60 min at 28 °C (Figure 9). For Ca^2+^, the reaction was extended to 5 h. The results showed that CaroS4K was most active with 1 mM concentrations of Ca^2+^ and Mg^2+^. Both ions showed a catalytic effect on CaroS4K’s ability to hydrolyze pMCL200 DNA. On the other hand, Cu^2+^ had no catalytic effect on the hydrolysis of pMCL200, even with varying concentrations. Most surprisingly, the Zn^2+^ buffer showed the most promising catalytic effect at a 0.1 mM concentration.

## 4. Discussion

The bacteriocin activity tests of the various *Pcc* strains showed that strain TO6 could produce LMWBs with a broad antimicrobial spectrum, inhibiting the growth of most tested strains. rif-TO6 was coupled with an *E. coli* 1830 strain with Tn5 to isolate approximately 10,000 Tn5 insertion mutant strains. Further screening resulted in 14 mutant strains which lost their secretion of LMWBs. TAIL-PCR experiments revealed that the same gene was disrupted by Tn5 insertion among the mutant strains. We focused on the DNA sequence of TT6-13. We obtained a 3724 bp long gene in eight TAIL-PCR experiments and identified two complete ORFs, designated ORF1 and ORF2. ORF1 was between the 690th and 3173rd base pairs and is highly homologous with the killer protein in *colicin* D157. ORF2 was located between the 3170th and 3439th base pairs. The results from the experiments indicated that ORF1 and ORF2 may be the killer and immunity proteins in the newly discovered LMWB. We named it the *carocin S4* gene, referring to the killer and immunity genes as *caroS4K* and *caroS4I*, respectively. Using DNAsis (Mac) software analysis, we found that the *carocin S4* gene, containing *caroS4K* and *caroS4I* within two open reading frames, is 2750 bp long. The 2484 bp long *caroS4K* gene can be translated into 827 amino acids, and the size of the protein is about 92 KDa, while the *caroS4I* gene, which can translate 89 amino acids, is 270 bp long, and the protein size is about 9.9 KDa.

The RT-PCR amplification results for *caroS4K* and *caroS4I* cDNA showed that the amount of cDNA in *caroS4I* is much larger than that in *caroS4K*, so we believe that the *caroS4K* and *caroS4I* genes use the same promoter when transcribing RNA. However, when *caroS4K* was disrupted by Tn5, *caroS4I* can still express its immune activity. This finding needs further research and investigation.

The amino acid sequence of CaroS4K was compared using BLAST through the database of the National Center for Biotechnology Information (NCBI). It was found that CaroS4K’s amino acid sequence is highly homologous to that of the S-type Pyocin domain-containing protein; hence, the CaroS4K structure was divided into four domains according to the structure of S-type Pyocin. Domain I (1-354th a.a.) is the first functional domain and is known as the receptor-binding domain. Domain II (355th to 460th a.a.) is the second functional area, whose function is still unknown. Domain III (461st to 677th a.a.) is the third functional domain and is known to serve as the translocation domain. Domain IV (678th to 827th a.a.), known as its killer domain, showed high sequence homology (80%) to the H-N-H endonuclease *Serratia odorifera*. 

The amino acid sequences of the *caroS31* structural genes were also homogenously aligned using BLAST, and the results showed 39% homogeneity with *Yersinia pseudotubercosis* IP 32953. Homology analysis of Carocin S4 with other LMWBs discovered in *Pcc*, Carocin S1, Carocin S2, and Carocin S3 was performed. The amino acid sequence of CaroS1K is 23% homologous with CaroS4K (Appendix A). Like CarosS4K, CaroS1K has been proven to exhibit DNase activity [34]. Comparison of the amino acid sequences between CaroS4K’s domain I and domain III with CaroS2K revealed a very high homology percentage of 90%, with a significant difference only in domain IV. We speculate that the two killer proteins recognize and deliver the killer domain into the target cells similarly but work differently in killing the target cells, as CaroS2K was proven to be an RNase [18]. The CaroS3K and CaroS4K amino acid sequences were found to be as much as 85% homologous (Appendix A). The two killer proteins cause the death of the target cell in a similar way, CaroS3K being a DNase [1].

The purified Carocin S4 contains two proteins, CaroS4K and CaroS4I, with molecular weights of about 90 kD and 10 kD, respectively. We hypothesized that when Carocin S4 is produced, it exists as a complex formed by the two proteins. The function of CaroS4I is to inhibit the activity of CaroS4K against the bacteriocin producer itself. The complex could not be completely separated during purification (Figure 5B). However, once purified, CaroS4K could inhibit the growth of the indicator bacterium 95F3 with a minimum inhibition concentration of 0.2 µg/mL (Figure 10).

Based on ICP-MS, the main metal ions contained in CaroS4K are Mg^2+^, Ca^2+^, Cu^2+^, and Zn^2+^. The ion concentration ratios of Mg^2+^, Ca^2+^, Cu^2+^, and Zn^2+^ in CaroS4K are 1: 0.7073, 1: 4.110, 1: 0.2060, and 1: 0.1550, respectively. Ca^2+^ may play an important role in folding the CaroS4K structure. As mentioned in the study on Colicin E7 and Colicin E7, it was found that the H-N-H motif of Colicin E7 is combined with a Zn^2+^, while the H-N-H motif of Colicin E9 can be combined with a Zn^2+^ or a Ni^2+^ [9,35]. Based on the results of this study, we speculate that the H-N-H motif of CaroS4K may be combined with a Ca^2+^.

The results of the reaction of pMCL200 DNA and CaroS4K imply that CaroS4K should be a non-specific endonuclease. Non-specific endonucleases are widely distributed among a variety of organisms and play an important role in DNA replication, recombination, repair, and cell division, as well as cell defense [36,37]. Non-specific endonucleases kill target cells by hydrolysis of the genomic DNA to increase their own capacity for survival. Moreover, several studies have mentioned that some non-specific endonucleases have both endoenzyme and exoenzyme activities [37]. CaroS4K may be the same as these non-specific endonucleases, with both endoenzyme and exoenzyme activities; however, further experimental evidence is required. CaroS4K hydrolyzes genomic DNA and plasmid DNA, unlike CaroS1K and CaroS3K, which can only hydrolyze genomic DNA [1,18].

The optimal reaction temperature for CaroS4K to hydrolyze dsDNA is approximately 50 °C. Studies on non-specific endonucleases show that most non-specific endonucleases are not thermally stable, and the ideal temperature for hydrolyzing nucleic acids is 30 to 70 °C. For example, the ideal temperature for ColE7 to hydrolyze dsDNA is about 50 °C [36]. Furthermore, we hypothesized that CaroS4K, like most nucleases, requires divalent metal ions to catalyze its DNase activity. CaroS4K hydrolytic DNA is less active without any binary metal ions. The metal ions Mg^2+^, Ca^2+^, and Zn^2+^ can hasten the hydrolysis of DNA. However, we speculate that the ion catalytic space size that may act on CaroS4K and DNA is better suited for Mg^2+^ and Ca^2+^. As mentioned in Shankar’s paper, non-specific endonucleases, like most restrictive endoenzymes and other endonucleases, require a reaction of divalent metal ions in the catalytic hydrolysis of DNA [37]. CaroS4K, like ColE9 [36], requires Mg^2+^ and Ca^2+^, with an optimal concentration of 1 mM for Mg^2+^ and Ca^2+^. Similarly, when the concentrations of Mg^2+^ and Ca^2+^ are too high (100 mM), the activity of CaroS4K is inhibited. 

The in vitro experiment on the inhibition of CaroS4I on the DNase activity of CaroS4K revealed that CaroS4I could not protect the indicator bacterium 95F3 from being killed by CaroS4K, nor inhibit CaroS4K from hydrolyzing the DNA. We hypothesize that, in vitro, CaroS4I is combined with CaroS4K’s translocation domain; hence, it does not affect CaroS4K DNase activity (Figure 11).

So far, it remains unclear how the process of DNA hydrolysis from non-specific endonucleases is different from the much-studied restriction enzyme DNA hydrolysis. Although many proteins with H-N-H motifs have been found, most of them have been deduced only by amino acid sequence comparisons. So far, only bacterial protein endonucleases have been used to explore the structural functions of H-N-H motifs with restrictive enzymes, but it is still unclear what role H-N-H motifs play in the hydrolysis mechanisms for DNA. It is hoped that further research on Carocin S4 will lead to the identification of the mechanisms for DNA hydrolysis from non-specific endonucleases and the role of H-N-H motifs in the protein hydrolysis of DNase.

## 5. Conclusions

A novel *carocin S4* gene was successfully isolated from *Pcc* strain TO6. The 2750 bp gene contains *caroS4K* and *caroS4I* within two open reading frames. It encodes the LMWB Carocin S4, which contains two proteins, CaroS4K (killer protein) and CaroS4I (immunity protein), with molecular weights of about 90 kD and 10 kD, respectively. CaroS4K exhibits a DNase activity similar to CaroS1K and CaroS3K but can hydrolyze both genomic and plasmid DNA. Temperature and the presence of the divalent metal ions Mg^2+^, Ca^2+^, and Zn^2+^ catalyze its DNase activity.

## Figures and Tables

**Figure 1 microorganisms-11-01854-f001:**
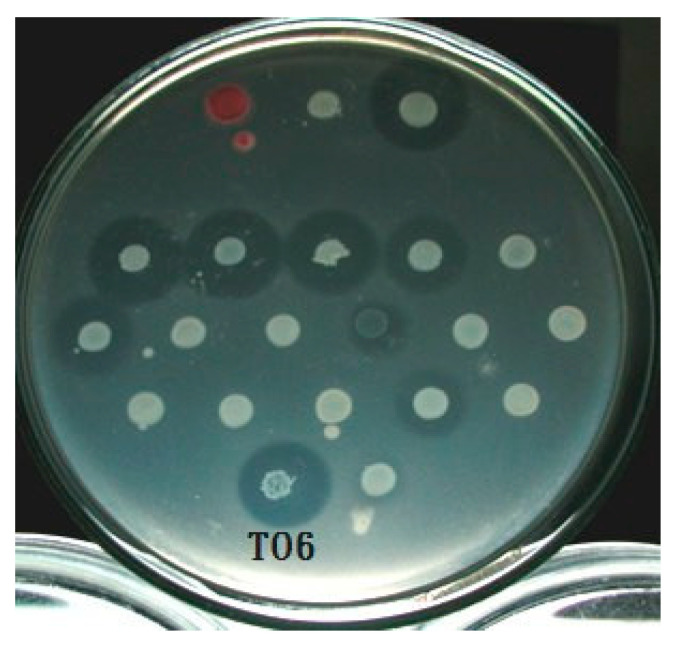
Bacteriocin assay. As a result of the cross-comparison, it can be seen from the figure that TO6 has a more obvious antibacterial ring.

**Figure 2 microorganisms-11-01854-f002:**
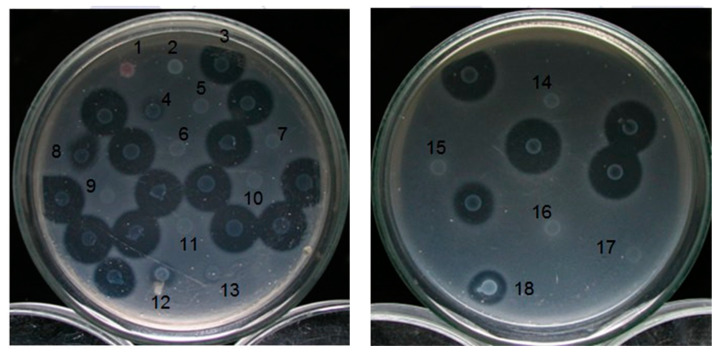
Bacteriocin assay for insertion mutants. *Pcc* strain 95F-3 was used as the indicator strain. 1: *Serratia sp.* (marker); 2: *Escherichia coli* 1830; 3: Rif-TO6; 4: TT6-2; 5: TT6-1; 6: TT6-3; 7: TT6-5; 8: TT6-4; 9: TT6-6; 10: TT6-12; 11: TT6-9; 12: TT6-24; 13: TT6-14; 14: TT6-13; 15: TT6-19; 16: TT6-21; 17: TT6-22; 18: TT6-25. The Tn5 insertion mutation strain, whose bacteriological proteins still functioned normally, is unmarked.

**Figure 3 microorganisms-11-01854-f003:**
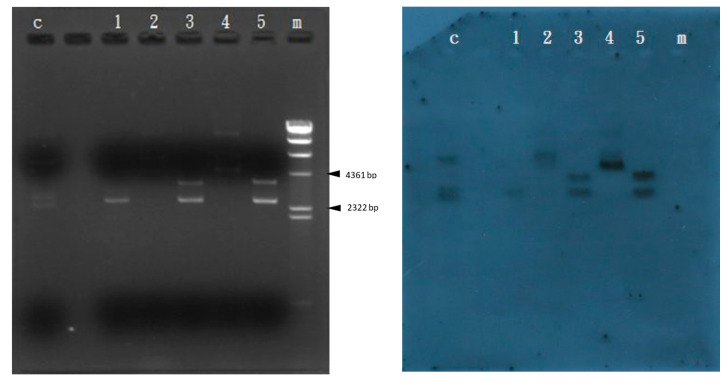
Results of the *carocin S4* Southern blotting experiment. C: control group; 1: pGEM-T Easy/*pvuII*; 2: pGS4KI-1; 3: pGS4KI-1/*pvuII*; 4: pGS4KI-2; 5: pGS4KI-2/*pvuII*; m: λDNA/*Hind*III marker.

**Figure 4 microorganisms-11-01854-f004:**
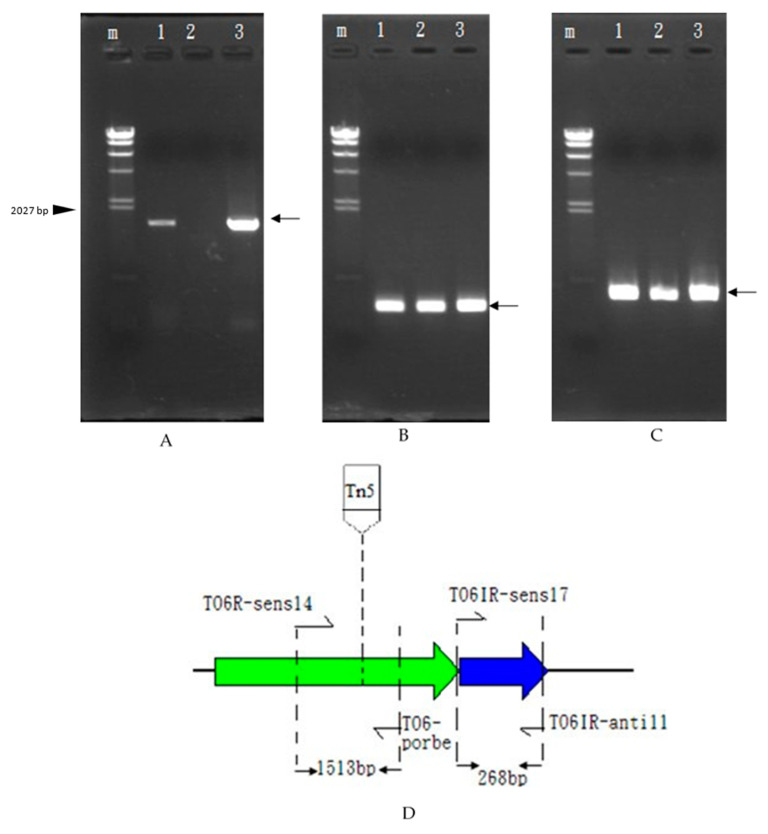
RT-PCR experimental results: electrophoresis map. The increase in the position is indicated by the arrow. 1: rif-TO6; 2: Tn5 insertion mutant TT6-13; 3: TT6-13/pGS4KI; m: λDNA/*Hind*III marker. (**A**) *caroS4K*, 1513 bp. (**B**) *caroS4I*, 268 bp. (**C**)16S rRNA, 397 bp. (**D**) RT-PCR diagram of *carocin S4*.

**Figure 5 microorganisms-11-01854-f005:**
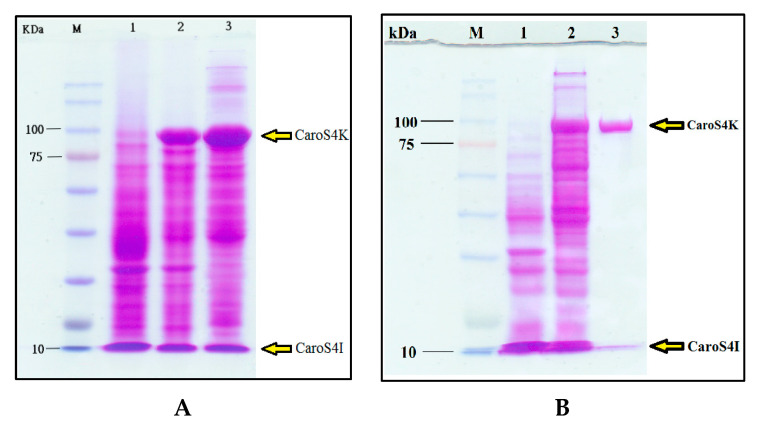
CaroS4K mass expression and purification. (**A**) After BL21/pETS4KI-2 was induced by IPTG, a large amount of CaroS4K was produced. (**B**) Purifying the crude protein solution containing CaroS4K revealed the two proteins, CaroS4K (90 kD) and CaroS4I (10.2 kD). Lane labels are as follows: 1: BL21/pET32a cell lysis, 2: BL21/pETS4KI-2 cell lysis, 3: BL21/pETS4KI-2 crude protein.

**Figure 6 microorganisms-11-01854-f006:**
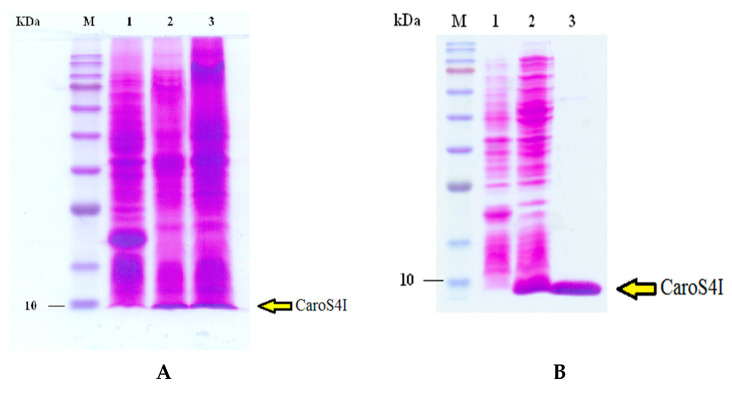
Mass expression and purification of CaroS4I. (**A**) SDS-page (12%) analysis showed that after BL21/pETS4I was induced by IPTG, a large amount of CaroS4I was produced. (**B**) The molecular weight of the purified protein was about 10.2 kD, which should be CaroS4I. Lane labels are as follows: 1: BL21/pET32a cell lysis, 2: BL21/pETS4KI-2 cell lysis, 3: BL21/pETS4KI-2 crude protein.

**Figure 7 microorganisms-11-01854-f007:**
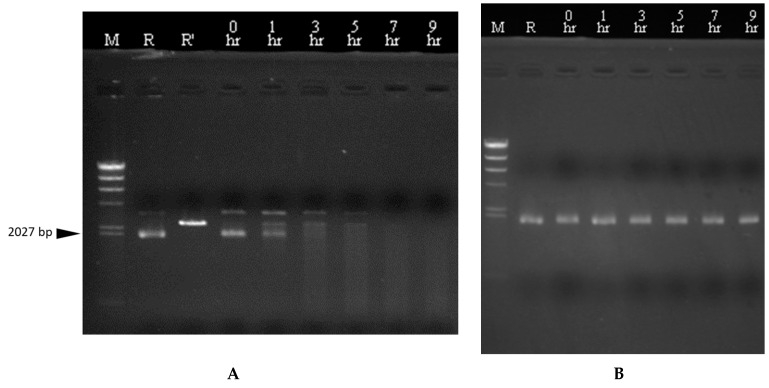
pMCL200 reacts with CaroS4K for different lengths of time. The 1 mM Ca^2+^ buffer containing pMCL200 (100 ng) was incubated with (**A**) and without (**B**) CaroS4K (1 μM) for different lengths of time at 28 °C. The reaction time was increased from 0 to 9 h, as indicated at the top of each lane. The lane labels are as follows: M: λDNA/*Hind* III; R: pMCL200 (100 ng); R’: pMCL200/*Sac*I.

**Figure 8 microorganisms-11-01854-f008:**
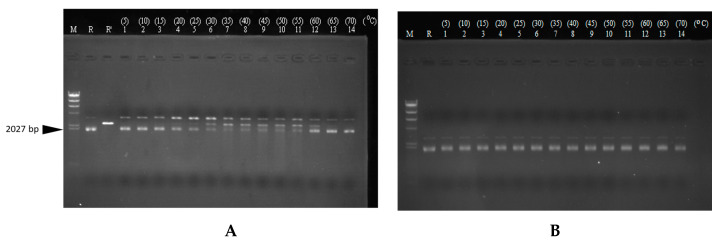
DNA from pMCL200 reacts with CaroS4K at different temperatures. The 10 mM Ca^2+^ buffer containing pMCL200 (100 ng) was incubated with (**A**) or without (**B**) CaroS4K (1 μM) for 60 min at different temperatures. The temperature was increased from 5 °C to 70 °C, as indicated at the top of each lane. M: λDNA/Hind III; R: pMCL200 (100 ng); R’: pMCL200/*Sac*I.

**Figure 9 microorganisms-11-01854-f009:**
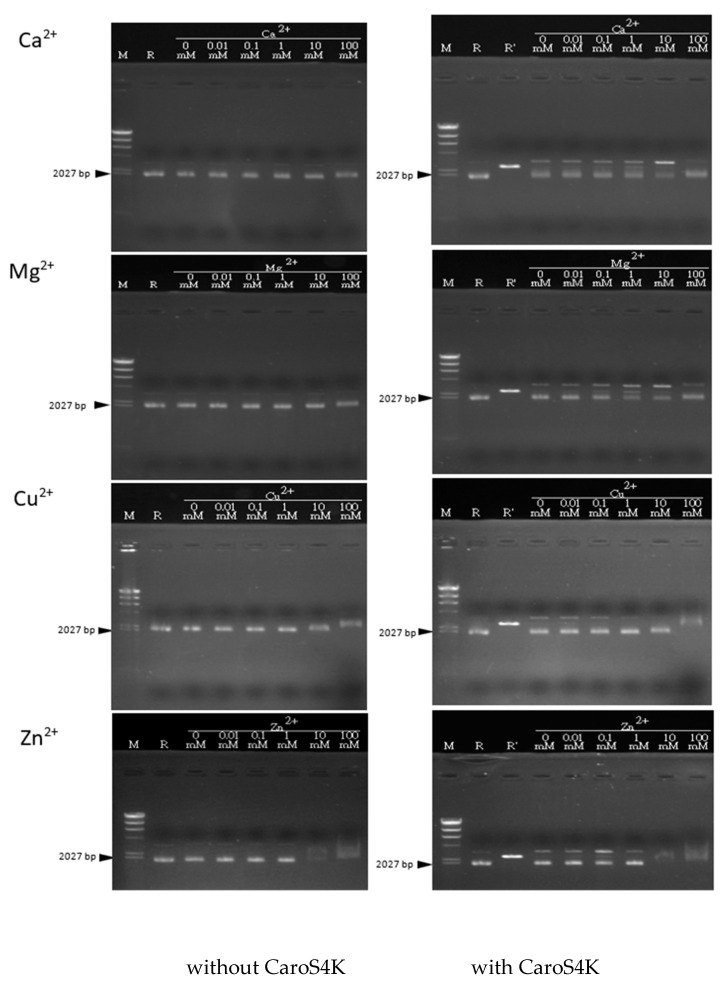
DNA from pMCL200 (100 ng) in different reaction buffers with or without CaroS4K (1 μM) for 60 min at 28 °C. The concentration of each divalent ion was increased from 0.01 mM to 100 mM, as indicated at the top of each lane. M: λDNA/*Hind*III; R: pMCL200 (100 ng); R’: pMCL200/*Sac*I.

**Figure 10 microorganisms-11-01854-f010:**
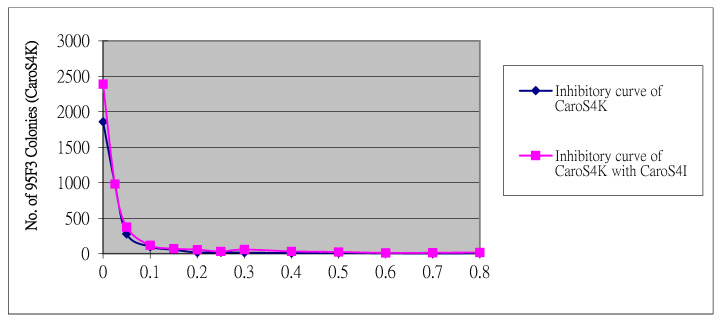
Minimum inhibitory concentration test for CaroS4K (indicator bacterium 95F3).

**Figure 11 microorganisms-11-01854-f011:**
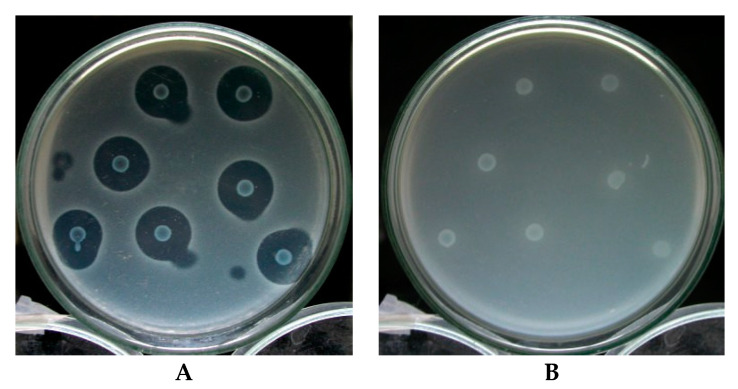
CaroS3I activity test using 95F3 (**A**) and 95F3/pGS3I (**B**) as indicator strains. All unlabeled colonies are rif-TO6.

**Table 1 microorganisms-11-01854-t001:** Metal ion contents in CaroS4K determined with ICP-MS.

Element	Mass	Measured Value of Blank (ppb)	Measured Value of CaroS4K (ppb)	CaroS4K: Metal
Mg	24.31	2.061	88.03	1: 0.7073
Ca	40.08	46.54	870.2	1: 4.110
Cr	52.00	<0.013	0.815	1: 3.135 × 10^−3^
Mn	54.94	<0.003	5.543	1: 0.02018
Fe	55.85	<0.094	11.08	1: 0.03968
Ni	58.69	<0.009	8.482	1: 0.03013
Cu	63.55	4.457	69.90	1: 0.2060
Zn	65.38	2.292	52.96	1: 0.1550
Cs	132.9	<0.001	0.035	1: 5.3 × 10^−5^
Ba	137.3	0.054	0.658	1: 8.80 × 10^−4^
Pb	207.2	0.067	1.464	1: 1.34 × 10^−3^

## Data Availability

The datasets used and analyzed during the current study are available from the corresponding author on reasonable request.

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
