# Peer review of "A Novel Deoxyribonuclease Low-Molecular-Weight Bacteriocin, Carocin S4, from Pectobacterium carotovorum subsp. carotovorum"

_microorganisms, 2023, doi:10.3390/microorganisms11071854_

Round 1
Reviewer 1 Report
The manuscript by Wu et al. deals with the characterization of a novel antibacterial activity that displays a DNAse activity. This bacteriocin turns out to be a 90 kDa protein. Therefore it cannot be regarded as a low molecular weight bacteriocin, as it is mentioned in the title.
The authors analyzed the sequences of both the gene that encodes the killer protein and the gene that encodes a hypothetical immunity protein and found that the killer sequence is similar to other nucleases already described. However, it is not correct to express how similar they are as percentage of homology or percentage of homogeneity as the authors mentioned in the manuscript.
The killer protein, CaroS4K, was expressed in Escherichia coli and the purified. The figure 5 shows the result of this procedure. The overexpression of this protein upon addition of IPTG is quite remarkable. However, the authors inferred that the immunity protein is expressed as a complex with the bacteriocin since they observed a strong protein band of 10 kDa, the expected molecular wight of CaroS4I. I think this is not correct since the same band is present in the control sample from bacteria bearing the empty vector, pET32a, which by the way does not contain the band of CaroS4K. If this complex really occurs, then it should be also present when CaroS4I is purified, yet CaroS4K is absent in the gel shown in figure 6.
Regarding the nuclease activity, if the authors only shows figure 7 in the main manuscript, one can conclude that CaroS4K may have a specific cut since there is no smear pattern of the DNA but rather define bands appear upon incubation with purified CaroS4K. However, the non-specific endonuclease activity is only apparent in figure 5 of the supplementary material where the typical smear can be observed from 3 h onward. This figure should be moved to the main manuscript. Moreover, the proposed mechanism is based on this figure.
Finally, the speculation about the promoter should be avoided since there is no enough data.
The English should be polished
Reviewer 2 Report
The manuscript "A novel deoxyribonuclease low-molecular-weight bacteriocin, 2
Carocin S4, from Pectobacterium carotovorum subsp. carotovorum " aims to characterize structurally and functionally a bacteriocin (and its gene) produced by Pectobacterium carotovorum. Although written in a straightforward manner, a general revision of the text should be performed. The paper is similar to others published by authors of the same group regarding other bacteriocins produced by Pectobacterium carotovorum.
Regarding specific parts of the text:
· Through-out the text the authors claim to have characterized a LMWB, or Low Molecular Weight Bacteriocin – as the authors state:”The 2484-bp long caroS4K gene can be translated into 827 376 amino acids, and the size of the protein is about 92 KDa”. What kind of criteria did the authors use to call this protein a Low Molecular Weight bacteriocin?
· Lines 130-137 - The verb tense used in this paragraph is different from that used in the rest of the text.
· Supplementary Figure 1 needs a legend. Interpretation is not straightforward.
· Figure 1 relative to the Bacteriocin assay - contrary to what is stated by the authors, it is not obvious that T06 has the strongest antibacterial ring. At least two other (non-identified in the figure, but positioned top-left) have the same or a even higher inhibition halo.
· Line 179 – “Dicer…purification.” Some context has to be given, otherwise the statement seems displaced and non-sense.
· Lines 355-357 – English must be reviewed. Zn has no catalytic effect - it affects the catalyzing effect of CaroS4K.
· Lines 381-383 – Is this sentence supported by the analysis of the gene cloned?
· Figure 8 – low quality image. Y axis: number of colonies? Is it exactly this what the authors what to mean? Because more than 2000 colonies are not countable on any plate. Or is it a concentration?
Reviewer 3 Report
In this manuscript, the authors discovered and characterized a novel bacteriocin, Carocin S4, from a plant pathogen Pcc by conducting genetic screens. The gene contains two ORFs and encodes a protein complex: a larger subunit (CaroS4K) servers as a toxin with deoxyribonuclease activity that requires specific metal ions and temperature; while a smaller subunit (CaroS4I) potentially serves as an antitoxin. The experiments were well-designed and performed, and the results were presented effectively. Overall, this manuscript is well-prepared, and the findings could significantly benefit the filed. To move the manuscript forward to publishable level, couple of issues need to be addressed:
1. Are there any homologs of the inhibitory protein CaroS4I across different species?
2. What’s the molecular ratio of CaroS4K and 4I in the purified complex?
3. The authors have realized that CaroS4I was unable to protect 4K in both DNA cleavage assay and bacteria killing assay. These results raise an important question: how does Carocin S4 distinguish the self and non-self DNA? It seems unlikely that CaroS4I interacts with 4K differently inside and outside of cells.
4. The metal ion content assay lacks sufficient description and discussion. How were the metals and proteins quantified and normalized? Since the “endogenous” metals were measured (there was no EDTA removal and metal re-coordination step), the presence of low-abundance species such as Cu and Zn could be due to background contamination from proteins purified along with CaroS4K. In this case, how to explain the molecular ratios of Mg and Ca? Is it truly possible for one CaroS4K to coordinated with 4 Ca ions?
5. There are two “Figure 8”, the later one should be “Figure 9”. It would be beneficial to determine the metal dependency in the same set of assays, using EDTA treatment as a negative control.
6. It would be worthwhile to explore Caro4K’s enzymatic activity using mutagenesis strategy.
7. In the DNA gels (e.g., Fig 3, 4, 7, 8) lack information regarding the molecular weight of the Marker.
8. The figure legends for Supplementary Fig. 1 and 2 are missing.
9. In Line 336, only “2+” should be superscripted, not “Ca”.
minor typo mistakes.
Round 2
Reviewer 3 Report
It is ok for publication